# The Importance of Nutrition in Menopause and Perimenopause—A Review

**DOI:** 10.3390/nu16010027

**Published:** 2023-12-21

**Authors:** Aliz Erdélyi, Erzsébet Pálfi, László Tűű, Katalin Nas, Zsuzsanna Szűcs, Marianna Török, Attila Jakab, Szabolcs Várbíró

**Affiliations:** 1Hungarian Dietetic Association, 1034 Budapest, Hungary; aliz.erdelyi@mdosz.hu (A.E.); zsuzsanna.szucs@mdosz.hu (Z.S.); 2EndoCare Institute, Endocrinology Center, 1037 Budapest, Hungary; tuul68@gmail.com (L.T.); drnaskatalin@yahoo.com (K.N.); 3Faculty of Health Sciences, Department of Dietetics and Nutritional Sciences, Semmelweis University, 1088 Budapest, Hungary; 4School of PhD Studies, Semmelweis University, 1085 Budapest, Hungary; 5Department of Obstetrics and Gynecology, Semmelweis University, 1082 Budapest, Hungary; varbiro.szabolcs@med.semmelweis-univ.hu; 6Department of Obstetrics and Gynecology, Faculty of Medicine, University of Debrecen, 4032 Debrecen, Hungary; ja@med.unideb.hu; 7Department of Obstetrics and Gynecology, University of Szeged, 6725 Szeged, Hungary

**Keywords:** menopause, perimenopause, dietetics, cardiovascular risk, healthy lifestyle, nutritional status, medical nutrition therapy, healthy eating guidelines

## Abstract

Menopause is associated with an increased prevalence of obesity, metabolic syndrome, cardiovascular diseases, and osteoporosis. These diseases and unfavorable laboratory values, which are characteristic of this period in women, can be significantly improved by eliminating and reducing dietary risk factors. Changing dietary habits during perimenopause is most effectively achieved through nutrition counseling and intervention. To reduce the risk factors of all these diseases, and in the case of an already existing disease, dietary therapy led by a dietitian should be an integral part of the treatment. The following review summarizes the recommendations for a balanced diet and fluid intake, the dietary prevention of cardiovascular diseases, the role of sleep, and the key preventive nutrients in menopause, such as vitamin D, calcium, vitamin C, B vitamins, and protein intake. In summary, during the period of perimenopause and menopause, many lifestyle factors can reduce the risk of developing all the diseases (cardiovascular disease, insulin resistance, type 2 diabetes mellitus, osteoporosis, and tumors) and symptoms characteristic of this period.

## 1. Introduction

As life expectancy increases, the number of menopausal women worldwide is increasing, with an estimated 1.2 billion worldwide by 2030 [1]. In a woman’s life, menopause can be said when menstruation no longer occurs within a year after the last period. This usually happens between 45 and 55 years of age. Menstrual irregularities, including amenorrhoea, can even last for years; such period is called perimenopause. Hormonal changes, which cause both somatic and psychological changes, begin even earlier, which is the period of menopausal change associated with the loss of fertility in women over 40 (‘menopausal transition’).

In the female body, estradiol is a hormone with extensive metabolic effects, so the absence of the cycle and the lack of periodic exposure to estrogen and progesterone cause changes in the target tissues of sex hormones, as well as the reproductive system. Estradiol affects the central nervous system, and increases food intake and basal energy consumption (basal metabolism). It increases gluconeogenesis in the liver, having an opposite effect to insulin. In skeletal muscles, estradiol increases insulin sensitivity and glucose uptake [2] and improves the function of pancreatic beta cells by increasing insulin secretion. With the onset of menopause and due to the lacking effect of estrogen, basal metabolism of the female body decreases significantly [3]. The hunger-suppressing effect of estrogen on estrogen alpha receptors in the central nervous system is also reduced, resulting in higher calorie intake [4,5]. Body composition changes parallel to the decrease in basal metabolism, as body weight increases, and fat distribution changes to an increased visceral fat, as mass increases [6]. The excess fat storage leads to larger adipocytes and tissue remodeling of visceral fat. Local growth factors are secreted, inducing adaptive angiogenesis, high metabolic activity, and oxygen consumption, resulting in excess production of free oxygen radicals. In response to structural damage, immune cells are recruited and accumulated in adipose tissue. The increased secretion of pro-inflammatory signaling molecules induces local and low-grade systemic inflammation [7]. This low-grade systemic inflammation plays a key role in accelerating vascular damage [8]. 

During menopause, the risk and occurrence of several chronic diseases increase in connection with the decrease in estrogen levels. These are cardiovascular diseases, tumors (especially hormone-sensitive breast cancer), insulin resistance, type 2 diabetes (T2DM), and osteoporosis, the risk of which can be reduced by lifestyle modification. Hormonal changes during perimenopause and menopause cause several characteristic symptoms. The strength, frequency, and tolerability of the symptoms can also be influenced by lifestyle. During menopause, hormone-sensitive breast cancer is more common in those whose menstruation started earlier than the average and ended later. Additional risk factors are if the woman has not given birth to a child or breastfed for at least 12 months [9].

During perimenopause and menopause, hot flashes and night sweats are common, as well as headaches and joint pains may become more frequent. Women become more irritable and emotionally volatile, and their ability to concentrate deteriorates. These symptoms can be of different intensity and frequency but all of them can be said to be influenced by lifestyle. Seventy-five to eighty per cent of women suffer from the symptoms caused by menopause, and they are more severe in 20–30% of women [10]. In this period of life, the change of lifestyle, even if it does not eliminate the symptoms and associated diseases, delays their development, makes them more bearable, and at the same time, makes everyday life easier [11]. In the case of obesity or overweight, losing just 5 kg of weight improves the tolerability of hot flashes by 30%. Regular exercise ensures metabolic health and reduces the number and intensity of hot flashes [12]. 

In a holistic approach to the care of healthy menopause, changes in the physiological processes, which occur as a result of the drop in estrogen levels, should also be followed by changes in the diet. The nutrition of the population of developed countries is still far from food-based dietary guidelines (FBDG) [13,14,15,16,17,18,19,20]. The frequency of obesity is increasing every year; between 1975 and 2016, the prevalence of obesity almost tripled. Based on WHO’s data from 2016, 40% of women are overweight and another 15% are obese [21]. Unhealthy Western nutrition is characterized by high fat (38% of energy) and salt intake (9.2 g/day in women in the age group of 50–59 years [22,23]), low vegetable and fruit consumption [24], and low calcium (730 ± 277 mg/day) intake [25].

Given all of this, significant steps must be taken with the help of dietitians to change the diet of those affected by menopause and perimenopause. All those diseases and unfavorable laboratory values, which are characteristic of this period of women, can be significantly improved by eliminating and reducing dietary risk factors. Menopause is associated with an increased prevalence of obesity, metabolic syndrome, cardiovascular diseases, and osteoporosis [26]. To reduce the risk factors of all these diseases, and, in the case of an already existing disease, dietary therapy should be an integral part of the treatment. Nutritional intervention is an essential element of the primary prevention of chronic diseases [27].

The European Menopause and Andropause Society (EMAS) also emphasizes the importance of lifestyle. There is growing evidence that lifestyle factors such as diet, physical activity, smoking, and alcohol consumption have a significant impact on health and menopausal symptoms. Regardless of menopause, women gain an average of 10 kg between the ages of 40 and 60. According to the recommendation of EMAS, women should be supported, among other things, in regular exercise and in creating a diet in accordance with national nutritional guidelines, to manage health risks, and prevent excessive body weight gain [28]. The following review summarizes the recommendations for a balanced diet and fluid intake in menopause, the dietary prevention of cardiovascular diseases, the role of sleep in the diet, and the key preventive nutrients in menopause, such as vitamin D, calcium, vitamin C, B vitamins, and protein intake.

## 2. Balanced Nutrition Recommendation

The quality of the diet is a determining element of an individual’s health at every stage of life. Its role in the development of chronic, non-infectious diseases has now been sufficiently proven. Therefore, regarding nutrition, the intervention of a dietitian is an essential element of prevention [27]. A healthy diet can help counter the symptoms of perimenopause and menopause and prevent many chronic diseases, such as cardiovascular disease, diabetes, and different types of malignancies. In the Epidemiology of Obesity prospective cohort study of 6671 individuals aged 45 to 65 years conducted in The Netherlands, dietary intake of fruits, vegetables, vegetable fats, and oils resulted in lower visceral fat mass. More than half of the 3576 women in the study were in menopause [29]. In Europe, recommendations for a balanced healthy diet are described in the Food-Based Dietary Guidelines (FBDGs) [13,14,15,16,17,18,19,20,30]. There are many diets which have a positive effect on chronic non-infectious diseases and weight management, such as the Mediterranean diet and the very low-calorie diet (VLCD-1200 kcal/day). Diets providing less than 1200 kcal/day may yield micronutrient deficiencies which could negatively affect not only the nutritional status but also the weight management outcome [31]. Therefore, low-calorie diets (LCDs) and VLCDs are used only in clinical practice. Balanced hypocaloric diets can be managed to individuals and may, therefore, have better chance for long-term success [31]. The other popular diet is the Mediterranean diet. The Mediterranean diet is characterized by foods with anti-inflammatory and antioxidant action. There is evidence that the Mediterranean diet affects weight management, blood sugar control, and cardiovascular diseases [5]. At the same time, Mediterranean meal planning is not sustainable in most the European countries. Summarizing the healthy, balanced diet is achievable in the long term, therefore it is preferred.

### 2.1. Nutritional Status during Perimenopause

During the period of perimenopause and menopause, due to the change in hormone levels (decrease in sex steroids), the basal metabolism of the female body decreases significantly, which can mean a decrease in the basal metabolic rate (BMR) of up to 250–300 kcal per day. In the case of an unchanged lifestyle, it may result in an annual weight gain of 2 kg [3]. Body composition changes parallel to the decrease in basal metabolism. During this period, the most common abnormal nutritional conditions are overweight and obesity. Weight gain, typically, means an increase in abdominal (visceral) fat mass [11]. In addition, due to the aforementioned hormonal changes, fat-free mass (FFM) and skeletal muscle mass (SMM) are characterized by a decrease, which can lead to sarcopenia and, in the case of two pathological body composition changes, sarcopenic obesity, in which, in addition to abnormal fat mass, muscle function and muscle mass decrease [10,32,33].

The prevalence of obesity increases with age. The incidence of abdominal obesity in women increases with age, and a rapid increase is observed in middle-aged women. Weight gain is a symptom of menopause, experienced by 60–70% of middle-aged women. On average, women gain about 6.8 kg per year during their midlife period (ages 50–60), regardless of their initial body size, race, or ethnicity [34]. According to 2016 WHO data, 55% of women are overweight or obese [21]. 

In menopause, due to the consequences of the hormonal changes, it is not sufficient to use the body mass index (BMI) combined with the waist circumference measurement, but it is necessary to use body composition analyzers based on the principle of bioelectrical impedance (bioelectrical impedance analysis, BIA), which are suitable for measuring body composition. Depending on the results obtained, the daily energy requirement and the ratio of the necessary nutrients can be determined. With individualized energy intake, in the case of weight reduction, the daily energy requirement cannot be less than the measured or calculated BMR. In the case of normal body weight, it can be determined depending on physical activity and BMR. During individualization of energy and nutrient intake in menopause, it must be taken into account that with the onset of menopause and due to the lack of the effect of sex steroids, the basal metabolism decreases. Even during menopause, the first step in dietary care is to determine the nutritional status. The easiest way to determine nutritional status is the body mass index (BMI in kg/m^2^, calculation: body weight in kilograms/height in meters squared). The mid-upper arm circumference (MUAC) provides an opportunity to estimate BMI [35]. BMI does not provide information regarding body composition. Additional measurement methods are needed to diagnose an abnormal nutritional status. In such cases, the waist–hip ratio, abdominal circumference, skinfold measurement, or performance scales are used to assess nutritional status. A more accurate measurement is provided by the use of body composition analyzers, which, in addition to calculating fat mass and body fat percentage, are suitable for determining fat-free body mass and skeletal muscle mass, as well as for detecting edema [36,37,38]. The following means overweight and obesity in women:Body fat percentage ≥ 30%;Abdominal circumference ≥ 88 cm (abdominal, visceral obesity);Waist–hip ratio ≥ 0.8 (abdominal, visceral obesity);MUAC ≥ 32 cm, then BMI ≥ 30 kg/m^2^ [39,40].

Sarcopenia can be diagnosed in women if the following conditions are met [41,42,43]: if the muscle mass: FFMI ≤ 15 kg/m^2^ (fat-free mass index);ASMI ≤ 5.25 kg/m^2^ (appendicular skeletal muscle index);

Or the muscle strength:
For BMI ≤ 23 kg/m^2^: ≤17 kgf;For BMI 23.1–26 kg/m^2^: ≤17.3 kgf;For BMI ≤ 26.1-29 kg/m^2^: ≤18 kgf;For BMI > 29.1 kg/m^2^: ≤21 kgf;Short physical performance battery (SPPB) SPPB ≤ 8.

### 2.2. Maintaining and Achieving a Healthy Nutritional Status

The symptoms of perimenopause and menopause can be significantly reduced and made tolerable by achieving and maintaining a healthy nutritional status. There is no single, ideal way to achieve a healthy nutritional status. An option must be chosen that allows the individual to change lifestyle in the long term and form a new system of habits. Complex, personalized lifestyle therapy has been proven to be more effective than individual therapeutic elements in themselves.

In reducing body weight, the energy requirement, including the energy requirement of basal metabolism (basal metabolic rate, BMR), is a determining factor. Energy intake below BMR does not lead to weight loss in the long term and is more difficult to maintain on a daily basis. Diets with an energy content of less than 1200 kcal/day are associated with a higher risk of micronutrient deficiency. In the case of low-energy diets, after one year, relapses can be observed in 45.6% of cases, and in the case of a very low-calorie diet (VLCD, <800 kcal/day), gallstones are common. Diets of 800–1000 kcal do not bring long-term lifestyle changes and sustainable weight loss [32,44].

During the dietary treatment of obesity, achieving a negative energy balance is of greatest importance. The ideal rate of weight loss is 0.5–1 kg of body weight loss per week, which occurs from fat body mass while maintaining muscle mass. This means a 15–30% or 500–1000 kcal lower energy intake than the current energy requirement. It approximately corresponds to an energy intake of 25 kcal/kg/day. The actual energy requirement is always calculated based on the current body weight [31,45,46,47].

To maintain or increase fat-free body weight and skeletal muscle mass, the daily protein intake should be 1–1.2 g/kg body weight (20% of energy), with regular exercise with free weights or against resistance. Diets with a high protein content (at least 20% of energy) only result in weight loss if the energy content is low. At the same time, ensuring the protein requirement (1–1.2 g/kg/day) is also essential for increasing and maintaining skeletal muscle [48]. Body weight loss usually slows down after 12 weeks, in which case the goal is to maintain the body weight achieved. It is recommended to achieve the clinically necessary weight loss during more than one cycle (one cycle means a certain amount of weight loss and maintenance). The average weight loss should be approximately 5–10% of the body weight [31,45,46,47].

In other respects, the diet should follow the guidelines of a balanced mixed diet. There is no difference in the number of kilos lost between low-carbohydrate diets (120 g CH/day during weight loss, 150 g/day during weight maintenance) and low-fat diets, but their cardiometabolic effects are different. Reducing energy by 500–700 kcal per day can be achieved primarily by omitting snacks between meals, incorporating small meals but reducing portion sizes, and avoiding sugar-containing liquids and alcoholic beverages [10,45,49,50].

### 2.3. Fluid Intake in Menopause

Adequate fluid intake is also extremely important during menopause, especially concerning cellular metabolism and maintaining the optimal functioning of hemostasis. It has an important role in regulating heat balance, detoxification, maintaining the proper functioning of the gastrointestinal tract and moisture of the mucous membranes, as well as turgor of the skin. *Adequate* daily *fluid intake* is important in the transport of nutrients and oxygen and contributes to the health of the skeletal system [26]. Estrogen and progesterone significantly affect not only the *cardiovascular system* but also fluid and electrolyte balance [51]. During menopause, hormonal changes affect the thirst as well, which may result in a significant decrease in fluid intake [52,53]. The individualized, appropriate amount of fluid intake is 33 mL/kg/day, which is recommended to be evenly distributed over the day [54,55,56].

## 3. Dietary Intervention of Chronic Diseases in Menopause

During menopause, the risk and occurrence of several chronic diseases increases in connection with the decrease in estrogen levels; therefore, the dietary therapy should be an integral part of the treatment and the nutritional intervention is an essential element of the prevention of chronic diseases [27].

### 3.1. Lipids Metabolism Disorders in Menopause

During perimenopause and menopause, the risk of cardiovascular diseases (CVD) increases with the decrease in estrogen levels [57]. It is a well-known fact that the population of women of reproductive age have a lower cardiovascular risk than the male population of the same age due to the protective effect of estrogen on the cardiovascular system [58]. In the presence of estrogen, the lipid profile is more favorable; the cholesterol level, LDL cholesterol fraction, and triglyceride level are also lower. With the onset of menopause, lipid parameters tend to deteriorate rapidly, the elasticity of blood vessels decreases, and the blood supply to the organs deteriorates. The increases in cholesterol levels after the onset of menopause already result in cholesterol values exceeding those of men of the same age group. In a few years, the worsening trend of LDL cholesterol and triglyceride levels catches up with, and even surpasses, that of the male contemporaries [59]. 

The risk of central obesity in menopausal women is five times higher than before menopause [26]. Epidemiological studies have shown that central obesity, dyslipidemia, glucose intolerance, and hypertension are the most common risk factors for CVD in menopausal women. The incidence of metabolic syndrome in postmenopausal women is 2–3 times higher than before menopause [60]. 

Estrogen exerts its protective effect before menopause by the following: acting on estrogen receptor alpha, estradiol increases the release of vasoactive compounds promoting vasodilation, nitric oxide, and prostacyclin, and shifts the renin-angiotensin axis towards the production of angiotensin 1–7 [61]. Estradiol also plays a role in the regulation of the vascular system with local anti-inflammatory effects and epigenetic modifying effects [61]. With the development of estrogen deficiency, the cardiovascular risk increases significantly. The increase in the risk is also influenced by the rate at which estrogen deficiency develops. In general, the greater the decrease in estradiol levels and the faster the rate of change, the greater the increase in cardiovascular risk [61]. The dietary treatment of cardiovascular diseases should aim to maintain and achieve a normal nutritional status, treat high blood pressure (one of the most important risk factors of which is currently the high salt intake), and manage unfavorable lipid profile changes. Strict adherence to a healthy diet can reduce the risk of cardiovascular death by 14–28% [62]. In the 2021 ESC Guidelines on Cardiovascular Disease Prevention in Clinical Practice (European Society of Cardiology, ESC), a Grade I/B recommendation is formulated; a balanced diet can be recommended to everyone to prevent cardiovascular diseases [63]. 

The fatty acid composition (quality) of the diet is more important than its total amount (grade B recommendation). The intake of saturated fatty acids (*SFAs*) may not exceed 10 E%. It is recommended to be achieved by replacing SFAs with polyunsaturated fatty acids (*PUFAs*) in the diet. Dietary intake of omega-3 fatty acids, including eicosapentaenoic acid (EPA) and docosahexaenoic acid (DHA), is extremely important (grade A recommendation). The fatty acid composition of the diet (saturated fatty acids, SFAs; trans-fatty acids, TFAs) affects serum cholesterol levels more than dietary cholesterol intake (grade B recommendation). In addition to all this, a dietary fiber intake of 30–45 g/day is recommended, mainly by consuming whole grains. It is recommended to consume at least 400 g of vegetables and fruits per day following WHO guidelines [64].

In the 2021 recommendation of the American Heart Association (AHA), balancing energy consumption is a priority. The guidelines encourage the consumption of fruits and vegetables in fresh, frozen, canned, and dried forms. Replacing refined grains with whole grains is associated with a lower risk of coronary heart disease. A higher intake of legumes and nuts and the consumption of 2–3 servings of fish per week are associated with a reduction in the risk of cardiovascular diseases. Liquid vegetable oils are recommended instead of tropical oils (coconut, palm, and palm kernel), animal fats (butter and lard), and partially hydrogenated fats. Saturated and trans fats (animal, dairy, and partially hydrogenated fats) should be replaced with non-tropical liquid vegetable oils. Evidence supports the cardiovascular benefits of dietary unsaturated fats, particularly polyunsaturated fats, primarily from vegetable oils [62]. 

Similar principles are implemented in the menopause diet recommendations of the British Dietetic Association. Considering the special target group, it recommends consuming at least four to five servings of unsalted nuts, seeds, and legumes per week and avoiding refined sugars, such as sweets, cakes, and soft drinks. It highlights the heart-friendly nature of oats, whole grains, and whole wheat bread, as well as legumes such as lentils, chickpeas, and beans, in addition to being excellent sources of fiber [10]. Individualized lifestyle changes are recommended for all patients with hypertension or with elevated but still normal blood pressure. The main elements of these lifestyle changes are weight control, reducing alcohol and salt consumption, and increasing calcium, potassium, and magnesium intake [64]. 

According to the European Society of Cardiology (ESC) and the European Society of Hypertension (ESH) guidelines, important elements of lifestyle modification are to reduce salt intake to less than 5 g per day (1A) while eating vegetables, fresh fruit, fish, seeds, non-saturated fatty acids (olive oil), and low-fat dairy products and avoiding red meat (Grade 1A recommendation) [65]. In the dietary prevention of cardiovascular diseases, the following should be emphasized in menopause: (a)Body weight control in menopause is recommended with energy intake corresponding to body composition measurements;(b)Salt consumption should be as close as possible to 5 g/day, preferring green and dried vegetable spices for seasoning;(c)The daily recommended intake of vegetables and fruit is 5 portions (500 g/day, of which 300–400 g of vegetables and 200–100 g of fruit): 3–4 portions of vegetables, 1–2 portions of fruit [13,14,15,16,17,18,19,20].

### 3.2. Carbohydrate Metabolism Disorders in Menopause

Metabolic changes during the menopausal transition include an increase in the proportion of adipose tissue, an increase in visceral fat, and a decrease in energy expenditure [6]. In addition, there is an impairment of insulin secretion and insulin sensitivity, as well as an increase in the risk of T2DM [66]. 

The changes affecting carbohydrate metabolism during menopause are the following [67]:In the absence of estrogen, the insulin secretion of the pancreatic beta cells decreases;Decreased insulin sensitivity in the muscles results in a decrease in glucose uptake;As a result of deteriorating insulin sensitivity in the liver, gluconeogenesis and lipogenesis increase, triglyceride accumulation increases, VLDL production increases, and insulin clearance decreases;As a result of the reduced insulin effect on adipose tissue, lipolysis increases, the size of fat cells increases, and inflammatory mediators accumulate;The resulting metabolic changes lead to the development of metabolic syndrome.

Therefore, the risk of developing a carbohydrate metabolism disorder is related to the presence or absence of sex hormones (menopause). A correlation can also be observed with the date of menopause. A large Chinese study revealed a clear correlation between the date of menopause and the risk of diabetes. The risk of developing a carbohydrate metabolism disorder was the lowest with menopause occurring between the ages of 45 and 49. Menopause at both younger and older ages was associated with an increased risk of diabetes. Menopause occurring before the age of 40 had the highest risk [68]. The risk of developing diabetes was increased by weight gain and unfavorable changes in body composition (adiposity) [69]. Based on these, it can be said that menopause is associated with an increased risk of T2DM. Lifestyle intervention, including diet and exercise, is the cornerstone of diabetes prevention and management [70].

According to the American Academy of Nutrition and Dietetics, for adults with prediabetes or type 2 diabetes, dietary therapy administered by a registered dietitian improves the effectiveness of medical treatment and longevity. Dietary therapy is more cost-effective, successful, and essential for prediabetes and obesity, and in preventing the progression of type 2 diabetes [71]. Lifestyle interventions, including dietary changes and regular exercise, aimed at moderate weight loss (5%) are the mainstay of treatment [72]. 

In perimenopause and menopause, if there is no impaired fasting glycaemia (IFG), impaired glucose tolerance (IGT), or T2DM present, it is important to achieve and maintain adequate nutritional status and body composition, as well as ensure energy, nutrient, and fluid requirements appropriate to age, nutritional status, physical activity, and diseases. Treatment of overweight/obesity is of particular importance in prediabetes and diabetes. Its elimination/reduction reduces the transition of pre-diabetes conditions to diabetes, reduces insulin resistance, improves glycemic control, reduces the number and dose of blood sugar-lowering drugs used in already existing diabetes, and reduces the cardiovascular risk accompanying these conditions. In the case of weight reduction, the daily energy intake is 500–700 kcal less than the requirement, but it cannot be lower than the BMR [73]. The daily recommended amount of carbohydrates in the diet should be at least 120 g, which is rich in vegetable fibers, vitamins, and minerals, contains only the necessary amount of added sugar, and is low in fat and salt. The WHO directive sets the recommended added sugar intake at no more than 10% of the daily energy intake, but an intake of less than 5% per day would have additional benefits in achieving a healthy lifestyle [74]. Based on the energy needs of an average adult woman, which means 2000 kcal, this is 200 kcal or approximately corresponds to a maximum of 50 g of sugar. Taking a ratio of 5%, this is 100 kcal, i.e., approximately 25 g of sugar [74]. Taking into account the reduction of the daily energy requirement, in the case of 1600 kcal, this means only 40 g and 20 g of added sugar corresponding to 160 and 80 kcal, respectively. This requires taste adjustment and better utilization of the natural sugar content of fruits and dairy products rather than replacing the missing amount or sweet taste with intense sweeteners and sugar substitutes.

Eating carbohydrates with a low glycemic index and rich in fiber has a health-protective effect. Regarding carbohydrate sources, preference should be given to vegetables, whole-grain foods, fruits, and dairy products without added sugar. Increasing the fiber content is beneficial since it slows down the absorption of carbohydrates. In addition, due to fibers’ satiating value, they improve the feeling of satiety and bowel function. Regarding fiber intake, the European guideline recommends an intake of over 25 g per day, 35–45 g for cardiovascular prevention [20,63]. The European Food-Based Dietary Guidelines (FBDG) recommend a minimum intake of sugar and fiber of 30 g [13,14,15,16,17,18,19,20]. A high-fiber diet has been shown to reduce the risk of obesity and has a protective effect against different diseases such as coronary heart diseases, intestinal malignancies, and type 2 diabetes. 

In the case of an already existing carbohydrate metabolism disorder, the adjustment of the diet is based on the energy and nutrient intake required to achieve or maintain normal body weight and on a defined amount and altered composition of carbohydrate intake [75]. This also includes determining the frequency and timing of meals. The individualized diet promotes the preservation of health as fully as possible with particular regard to the achievement and maintenance of individual goals related to nutritional status, provision of individual glycemic target values, and prevention or delay of complications. Establishing a daily eating schedule is an important criterion. A diet divided into several parts per day with defined carbohydrate content helps to limit the blood-sugar-raising effect of meals [75]. Efforts should be made to exclude added sugars from the diet. Taking into account the amount of fast-absorbing carbohydrates found in milk, fruits, and fermented (acidified) natural products and the time of consumption, they can be included in the diet [47]. In the case of already developed T2DM, adequate dietary therapy is an integral part of the toolset of treatment and self-management. Learning it under the guidance of an experienced dietitian can result in a 0.3–1.0% HbA1c reduction in type 1 diabetes and 0.5–2.0% in type 2 diabetes [76]. 

Most people with type 2 diabetes are overweight. In addition to ensuring the ratio of dietary components and aspects of healthy nutrition, it is necessary to limit energy intake in these cases. In addition to improving glycemic control, continuous and non-forced weight loss can contribute to regulating blood pressure and blood fat levels. In all stages of type 2 diabetes, it is fundamentally important to be aware of the energy content of food and follow and calculate carbohydrate intake. Regarding carbohydrate sources, preference should be given to vegetables, whole-grain foods, fruits, and dairy products, while foods and beverages containing added fat, sugar, or salt should be avoided. The dietary fiber ratio for people living with diabetes must be at least the same as the amount recommended for non-diabetic patients. Incorporating 30 to 50 g of dietary fiber into the daily carbohydrate intake is recommended. Increasing the fiber content is beneficial since it slows down the absorption of carbohydrates. In addition, due to its satiating value, it increases the feeling of satiety and improves bowel function [77]. 

Increasing fiber intake reduces the risk of developing metabolic syndrome. An increase in fiber intake of 10 g per 1000 kcal reduced the risk of metabolic syndrome by 0.1 unit [78]. By increasing fiber intake, the composition of the intestinal flora changes. The changes have a positive effect on metabolism, insulin sensitivity, and insulin secretion. The increased fiber intake can have a positive influence on bacterial flora similar to isoflavonoids (e.g., soy), pre-, pro-, and postbiotics [79]. The intestinal flora also play a role in estrogen metabolism. Bacteria with beta-glucuronidase activity can increase the level of biologically active estrogen in the body, thus slowing the development of estrogen deficiency and reducing the symptoms accompanying menopause. Based on previous research, the functioning of intestinal bacteria with beta-glucuronidase activity in addition to a diet rich in fiber does not represent an increased risk of breast cancer [80]. When formulating a fiber-rich diet, gradualness is important so the body can adapt to the changed diet. Excessive fiber intake, consuming more than 50 g of fiber per day, may have a negative effect because it may cause bloating, increase the binding and excretion of useful substances, and entail digestive problems. In each case, it is advisable to determine the dietary ratio of fats and the composition of foods containing fats on an individual basis, taking into account the cardiovascular risk. In diabetes complicated by dyslipidemia, it may be beneficial to consume 1.6–3.0 g of plant stanols and sterols per day. The sterols found in plants reduce the absorption of dietary cholesterol by binding to their receptors. The consumption of foods enriched with phytosterols is, therefore, favorable by reducing the total and LDL cholesterol levels of the serum. Natural ingredients should be preferred over foods and beverages containing added sugar. If necessary, energy-free and low-energy sweeteners can be used to replace added sugar, as they can reduce daily carbohydrate and energy intake [47]. 

The organizations performing scientific evaluation and risk assessment (EFSA in Europe and FAO/WHO JECFA on a global level) determined the acceptable daily intake (ADI) amounts regarding intensive sweeteners and sugar alcohols [81]. All additives approved before 20 January 2009 must be re-assessed. EFSA is conducting a consultation on assessing consumer exposure to sweeteners to develop a protocol. Even for the amount of sweeteners used to replace sugar, the recommended maximum is approximately 5% to 10% of the total daily energy intake, similar to that recommended in prevention [20].

### 3.3. Bone Metabolism Changes in Menopause

Osteoporosis is a chronic, progressive health problem that affects most women during menopause and has serious consequences. Osteoporosis is particularly common during this period in those with low serum vitamin D levels [11]. The average annual rate of bone loss during menopause, beginning 1–3 years before menopause and lasting 5–10 years, is approximately 2% resulting in an average 10–12% decrease in bone mineral density (BMD) in the spine and hip.

Among nutritional factors, bone health can be adversely affected by abnormal nutritional status (overweight, obesity, malnutrition, or sarcopenia), vitamin D deficiency, hypercalciuria, and malabsorption disorders (e.g., celiac disease, inflammatory bowel diseases, gastrectomy, chronic liver diseases, or gastrointestinal malignancies).

A distinction must be drawn between osteoporosis (low BMD) and risk factors for fractures. The causes of low BMD are menopause itself, age, genetic predisposition, abnormal nutritional status, other diseases, and medications that affect bone absorption and metabolism. Risk factors for bone fractures are low BMD, previous fractures, multiple fractures occurring in the family, old age, frailty syndrome, and other diseases (e.g., those associated with dizziness and general weakness) [82].

Based on the above-mentioned factors, it is necessary to introduce lifestyle changes during this period to reduce the risk of fractures caused by osteoporosis, which include maintaining/achieving a healthy nutritional status and balanced nutrition focusing on adequate intake of vitamin D and calcium, regular exercise, smoking cessation, and stopping alcohol drinking [11,82].

### 3.4. Cancer Prevention during Perimenopause and Menopause

In connection with female hormonal changes during perimenopause, the risk of developing hormone-sensitive breast cancer may increase, so in this period, in addition to the previously mentioned diseases (CVD, IR, T2DM, and osteoporosis), tumor prevention lifestyle changes (exercise, nutrition, avoidance of alcohol, tobacco, and coffee) are also important. 

During menopause, an increase in body weight, including visceral fat weight, is characteristic, which not only increases the risk of CVD but also the risk of IR and, through this, the development of tumors [11]. 

The most common cancer in women during menopause is breast cancer, for which the three leading risk factors are overweight (or obesity), regular alcohol consumption, and a sedentary lifestyle. A weight gain of 20 kg in adulthood doubles the risk of breast cancer. Regular alcohol consumption and a sedentary lifestyle in both the pre- and postmenopausal periods increase the risk of developing breast cancer [9]. Maintaining and achieving a normal nutritional status and body composition are also paramount in reducing the risk of cancer. In particular, the goal is to maintain and achieve a normal range of fat-free mass, and skeletal muscle mass. In addition to following the guidelines of a balanced diet, the regular inclusion of cruciferous vegetables in the diet (cabbage, broccoli, cauliflower, radishes, and canola) and the consumption of at least half a kg of vegetables and fruits should be highlighted. 

In the case of an already developed breast cancer, the focus is on maintaining normal nutritional status and skeletal muscle mass. In addition to cancer treatments, the diet is the same as the IR diet. In case of adequate energy intake or overweight or obesity, a 500–700 kcal less energy intake and a fat intake of 30–35% of energy are recommended, most of which consists of monounsaturated and polyunsaturated fatty acids, i.e., vegetable fats. Tumor size can be reduced with an optimized energy intake or in the case of obesity or overweight, with a 5–10% reduction in body weight [9].

### 3.5. The Role of Micronutrients in Menopause

Most of the vitamin D requirement is synthesized endogenously in the summer months. However, when there is not enough UV-B radiation, dietary vitamin D intake is also important (e.g., in Central Europe from October to March, or if someone does not spend enough time outdoors, uses sun protection creams, or wears clothing that fully covers the skin). With ageing, the rate of hydroxylation of vitamin D precursors in the body decreases, so the importance of exogenous vitamin D intake increases with age [83,84]. 

Vitamin D is a fat-soluble vitamin. Its sources are egg yolks, dairy products, offal, and foods supplemented with vitamin D. About 80% of dietary vitamin D is absorbed in the small intestine. If UV-B radiation is insufficient, routine dietary supplementation is required. Vitamin D supplementation is especially important for infants, young children, and the elderly population. Vitamin D preparations should be taken with meals [20]. Clinical studies have proven that osteoporosis treatments only achieve their effectiveness with adequate vitamin D supplementation (more than 1000 IU per day). Without vitamin D supplementation, the fracture risk reduction effect of osteoporosis therapies can decrease by up to 30%. From October to March in Central Europe, the UV-B radiation is so low that not enough vitamin D can be synthesized in the skin. Therefore, continuous vitamin D supplementation is recommended in these months for preventive purposes with a dose of 2000 IU per day. For people who do not have permanent access to sunlight, a year-round supplementation is recommended [85].

The calcium requirements can be covered by a balanced diet. In the case of a diagnosed deficiency, the appropriate intake can be ensured primarily by modifying the diet. Routine calcium supplementation is not recommended due to its cardiovascular risk [82]. The absorption of calcium is primarily influenced by vitamin D status, but it can be improved by the acidic components of dairy products and food matrix. Dairy products contain the most easily absorbable calcium, but consumption of mineral waters with high mineral content or hard water also contributes to ensuring the need. Regular consumption of soft-boned fish (eaten with bones), such as canned sardines, pickled herring, and anchovy, as well as oilseeds and foods supplemented with calcium, also contribute to calcium intake [86]. 

In addition to insufficient vitamin D status, the factor inhibiting calcium absorption and utilization is a diet rich in protein, dietary fiber, and phytates [20]. Supplements should only be used if a calcium deficiency can be diagnosed. The recommended daily calcium intake (recommended dietary allowance, RDA) for menopausal women (from the age of 51) is 1000–1200 mg [82]. The calcium intake recommended by the European Food Safety Authority (EFSA) is 950 ug over the age of 25 [20].

The protein intake should be 0.8 g/kg body weight/day following a balanced diet, and 1–1.2 g/kg body weight/day protein intake is recommended for those who exercise regularly and in case of weight loss, half of which should come from vegetable sources. A diet rich in protein (1.5–2 g/kg body weight/day) increases the risk of fractures [82]. 

Vitamin C is necessary for bone formation due to its role in collagen formation. Its absorption is about 80% if the daily intake is 100 mg/day. RDA is 100 mg/day, which can be provided through diet. Its sources are freshly eaten vegetables and fruits, especially peppers, currants, citrus fruits, and sauerkraut [20].

All menopausal women, not just those at risk of fracture, should be encouraged to adopt a bone-friendly lifestyle. This includes optimizing calcium and vitamin D status, getting adequate physical activity, and avoiding smoking and alcohol [87]. Prevention of osteoporosis begins in childhood and adolescence, as this is the age when peak bone mass is formed. The two pillars of prevention are maintaining adequate calcium and vitamin D status, as well as regular exercise. According to the American guideline, phytoestrogens, including isoflavonoids, can only have a mild effect on estrogen deficiency, thus, on the loss of bone minerals; therefore, they do not prevent the development of osteoporosis [82]. 

B vitamins also play an important role in menopause. They play a fundamental role in the processing of carbohydrates and the functioning of the nervous system [88,89]. Adequate B vitamin intake significantly reduces the serum homocysteine level and, in parallel, the risk of stroke [90]. High homocysteine levels have also been associated with osteoporosis and increased risk of bone fractures [91,92,93,94,95,96,97].

In the prevention and treatment of cognitive dysfunction and cognitive decline, which are common complaints in menopause, an *adequate supply of B vitamins* is extremely important [54,98,99,100,101,102,103]. A balanced B-vitamin intake, therefore, plays an important health-preserving role in menopause. A microbiome with the appropriate biodiversity and composition also contributes to this [104,105].

### 3.6. Soy and Phytoestrogens and Menopause

During perimenopause, women in question try many dietary practices. Perhaps the most common and at the same time the most controversial one is the consumption of soy and other phytoestrogens in the diet or as dietary supplements. Many studies for and against the topic can be read. While the isoflavone content of soy can have a positive effect on the strength and frequency of symptoms, phytoestrogens can have a negative effect on the treatment of hormone-sensitive breast tumors.

Menopausal hot flashes are rarer in countries where regular soy consumption is a part of the diet. The isoflavone content of soy foods may be effective in reducing menopausal symptoms. One study recommended 20 mg/day of soy isoflavones supplementation during perimenopause for symptom reduction. This corresponds to 400 mL/day of soy drinks or 80 g/day of other soy products (tofu, tempeh, or fermented soy products) [106]. According to another study, intake of more than 42 mg/day of soy isoflavones may have a tumor-reducing effect [107]. According to a third publication, regular soy consumption does not increase the risk of developing breast cancer [108]. However, the effectiveness of anti-estrogen therapy (e.g., Tamoxifen) can be reduced by regular soy consumption (examined substance: Genistein). However, according to an Asian population-level study (a study of isoflavones), consuming 10–15 g of soy protein (equivalent to 250 mL of soy drink) with a balanced diet and a healthy lifestyle can be safe even in these cases [9,109,110,111].

In summary, it can be said that there is no consensus among the scientific community on the effect of dietary soy on breast cancer and its treatment. It is important to point out that the above safe intake applies only to soy foods included in the diet and not soy isoflavones taken as dietary supplements [112].

## 4. The Role of Microbiom in Menopause

Perimenopause and menopause are often associated with dysbiosis and gastrointestinal tract complaints. Estrogen levels can effect gut microbiota [113,114] and microbiome, in turn, can effect serum estrogen levels as certain microbes in the gut (called estrabolome) secrete beta-glucoronidase, a bacterial enzyme that deconjugates estrogens and phytoestrogens into their active forms which can be reabsorbed in the intestine and enter the bloodstream [115]. Estrabolome can be reduced by dysbiosis and, consequently, lead to further loss of circulating active estradiol metabolites [116]. This suggests that composition of the microbiota could determine the onset or progression of some menopause-related clinical conditions [117].

Probiotic supplementation in menopausal women suggest a favorable effect on some cardiovascular risk factors [118] through maintaining the integrity of the intestinal barrier, therefore, reducing translocation of bacteria through the gut wall and decreasing systemic inflammation. Fermentation of polysaccharides and undigested proteins by some microbial strains generates SCFA (i.e., acetate, propionate, and butyrate), possibly benefiting several metabolic pathways. Newest studies suggest that supplementation with probiotics improves cardiometabolic risk factors in postmenopausal women [119]. Maintaining healthy gut biom by a balanced diet according to the needs of menopausal women might help prevent dysbiosis.

## 5. Sleep and Menopause

Sleep difficulties are more likely to occur during menopause, with a self-reported rate of between 40% and 56%. For some women, sleep problems are severe, leading to chronic fatigue, affecting quality of life, and can have long-term consequences for mental and physical health [120]. In a meta-analysis, sleep disturbances during menopause were found to be common and significant. According to the results of the analysis, the general prevalence of sleep disorders based on these studies was 51.6% [121]. Deviation from the recommended 7–8 h of sleep in adults is associated with a higher risk of mortality and cardiovascular events [122]. Those who sleep less than 5 h a day are at the greatest risk of developing cardiovascular diseases. Those who sleep less than 7 h are also at increased risk of cardiovascular diseases and mortality [123]. Supporting this, in 2022, the American Heart Association increased the number of basic factors necessary to maintain optimal cardiovascular health from seven to eight, with the inclusion of the right amount and quality of sleep [124]. 

The circadian rhythm plays an important role in the regulation of metabolic processes. Sleep deprivation itself affects energy intake, glucose uptake, and leptin resistance. Sleep and circadian factors affect appetite, nutrient absorption, and metabolism. Disturbances in sleep and circadian rhythms can also worsen digestive disorders [125,126]. Sleep deprivation is associated with certain chronic diseases (e.g., cardiovascular diseases, diabetes, and tumors) and also increases the risk of becoming overweight [127]. 

Foods containing melatonin can directly affect sleep. Adequate amounts of tryptophan, a precursor of melatonin, have a positive effect on sleep. Thus, the different tryptophan content of foods can also cause melatonin levels to fluctuate. Improvements in sleep parameters (e.g., increased sleep time) were observed under the influence of tryptophan. It has been found that the quality of sleep deteriorates in case of reduced intake. Vitamins and trace elements participating as cofactors in melatonin synthesis are the following: folic acid, vitamins B6 and B12, magnesium, and zinc [69]. Adequate intake of these is also necessary for good sleep. 

Diet affects melatonin levels not only through tryptophan intake but also through raw materials containing melatonin in its natural form. In recent decades, melatonin has been widely identified in various foods. Among animal sources, eggs and fish have a higher melatonin content, while nuts have the highest melatonin content among plant-based foods. Among fruits, cherries, strawberries, and wine grape skins contain the highest amount of melatonin [128]. Research carried out with cherry varieties has shown an exceptionally high melatonin content in cherries, which suggests that cherries can be a natural source of melatonin and become suitable for the development of melatonin-rich functional foods [129]. Sources of the highest tryptophan content are not typical ingredients of the Western diet (just for information: the fatty meat of seals, their kidneys, and the beluga whale) [130]. Sea fish (codfish) and shellfish, algae (including spirulina), eggs (especially egg whites), soy, sesame seeds, sunflower seeds, pumpkin seeds, cheeses, and yeast contain high levels of tryptophan (see Table 1) [131]. 

A regular eating schedule plays a key role in maintaining the circadian rhythm. Irregular eating disrupts the biorhythm, thereby causing a shift in the sleep cycle and deterioration of sleep quality. Sleep is most affected by the timing and quality of dinner. It is recommended to have dinner at least 2 h before bedtime. Drinking a lot right before bedtime is not recommended. The daily fluid intake should be spread evenly throughout the day. Drinking water is best for thirst quenching.

A balanced diet has a positive effect on the quality of sleep. Foods and meals containing sufficient protein, carbohydrates, and fat are essential for good sleep quality. Not only the quantity of nutrients but also their quality is important. Scientific evidence points to the role of omega-3 fatty acids, which can positively influence the regulation of serotonin secretion. Consuming carbohydrates with low glycemic index, low glycemic load, and high fiber content is recommended [123]. In addition to all this, proper sleep hygiene and sleep self-examination also can help a lot. The use of various smartwatches and smart devices can also help to recognize and monitor sleep disorders, as well as to assess the effectiveness of the applied diet therapy and medical treatment [132,133].

## 6. Conclusions

Table 2 summarizing the main nutritional indications in menopause.

In summary, during the period of perimenopause and menopause, many lifestyle factors can reduce the risk of developing all the diseases (CVD, IR, T2DM, osteoporosis, and tumors) and symptoms characteristic of this period. The main messages of the recommendation are as follows:Achieving/maintaining a healthy nutritional status (BMI = 18.5 − 24.9 kg/m^2^, normal range of fat mass, and skeletal muscle mass) is the goal even during the perimenopause period;In case of overweight or obesity, an energy intake less than the current energy requirement (reduced by 500–700 kcal/day) and a protein intake of 1–1.2 g/day are recommended while following the recommendations of a balanced diet. Energy intake below BMR is not recommended in the long term;During perimenopause, the process of dietetic care for women should be based on the *Nutrition* Care Process Model (NCPM);The use of body composition analysis tools is ideal for assessing nutritional status;Following the guidelines of a balanced diet reduces symptoms and preserves health. Ensuring energy, nutrient, and fluid requirements appropriate to age, nutritional status, physical activity, and diseases are required as follows:
Establishing a physiological eating schedule is necessary;Simple, fast-acting sugars should be avoided;Protein intake should be 0.8–1–1.2 g/kg/day, half of which should come from plant sources;Adequate intake of calcium, vitamin D, vitamin C, and B vitamins is important;Adequate intake of n-3 LCPUFA and omega-3 fatty acids is necessary;Sugary and alcoholic beverages should be avoided;Fruits and vegetables provide vitamins, minerals, fiber, and other plant nutrients, such as antioxidants, to help protect the heart. The recommended daily intake of vegetables and fruits is 5 portions (500 g/day: 300–400 g of vegetables and 200–100 g of fruit), i.e., 3–4 portions of vegetables and 1–2 portions of fruit;Eating legumes (beans, peas, lentils, chickpeas, or soy) at least once a week is recommended;Regular consumption of low-fat protein sources (e.g., poultry, low-fat dairy products) helps to cover calcium needs;Moderate consumption of red and processed meats is recommended;Consumption of no more than 350–500 g boiled/steamed/fried (500–700 g of raw meat) red meat (e.g., beef and pork) per week is recommended. Intake of processed meat products should be only occasional, in small quantities. Incorporating at least one meat-free day per week can be useful. Meat can be replaced with fish, eggs, dairy products, and the right combination of legumes, grains, and nuts;Moderate consumption of fats and sweets is important. Consumption of vegetable fats and occasional consumption of high-fat foods is recommended. Sunflower oil for frying, and olive, rapeseed, linseed, soybean oil, etc., as salad dressing are recommended;Using as little amount of sugar and salt as possible to flavor food and drinks is important. A portion of salt can be replaced with fresh or dried herbs;At least two servings per week (100–120 g/occasion) of deep-sea fish with fatty meat (e.g., consumption of salmon, mackerel, tuna, herring, and sardines) or freshwater fish (e.g., trout and silver carp) is recommended;Consumption of 30 g of unsalted nuts, other oily seeds, or seeds per day *can be beneficial*. When it comes to frequency, it is important to take body weight into account;Incorporation of foods and ingredients with a higher fiber content daily is recommended: whole grain bread, fiber-rich breakfast cereals without added sugar, and brown rice. Oats, whole grains, whole wheat bread, and legumes such as lentils, chickpeas, and beans are excellent sources of fiber. The daily amount of dietary fiber should be 30–45 g, preferably mainly whole grains. One-third of the amount of grain consumed should be whole grains;The amount of saturated fat should not exceed 10% of the total energy intake. Replacement of saturated fats with monounsaturated and polyunsaturated fatty acids, or carbohydrates from whole grains is recommended. The amount of TFA should be reduced to the smallest possible amount so that the consumption of processed products is limited and the natural TFA intake is kept below <1E%;Eighty per cent of salt intake comes from processed foods, and only 20% is consumed in the form of added salt. It is recommended to reduce the amount and frequency of processed food consumption. Salt consumption should be as close as possible to 5 g/day, preferring fresh and dried vegetable spices for seasoning;Consuming dairy products corresponding to the calcium content of half a liter of milk per day is recommended. It is necessary to introduce lifestyle changes during this period to reduce the risk of fracture associated with osteoporosis. Changes should include maintaining/achieving a healthy nutritional status and balanced nutrition focusing on adequate intake of vitamin D and calcium, regular exercise, as well as quitting smoking and stopping alcohol drinking. Dietary supplementation of calcium and vitamin D should be considered based on the season and daily intake, as well as on the presence of osteoporosis and cardiovascular risk factors;A smoking-free lifestyle is recommended;Regular physical activity is essential.


## Figures and Tables

**Table 1 nutrients-16-00027-t001:** Tryptophan content of foods [131].

Food/Raw Material	Tryptophan Content (mg/100 g)
Cheddar cheese	574
Tuna	313
Salmon	285
Oat flakes	234
Buckwheat	192
Tofu	235
Kidney beans	104
Pumpkin seeds	576
Sunflower seeds	295
Almond	209
Peanut	193
Lamb (shoulders)	415
Chicken breast	404
Ground pork, lean	326
Turkey breast	287
Egg, whole	167

**Table 2 nutrients-16-00027-t002:** The main nutritional indications in menopause.

*Body composition*
Use body composition analysis tools to assess nutritional status Keep the weight in healthy range with adequate nutrients intake
Manage overweight, obesity: reduce current energy by 500–700 kcal/day Regularly physical activity
*Dietary recommendations*
Protein—0.8–1–1.2 g/bwkg/day
Calcium, vitamin D, vitamin C, vitamin B
n-3 LCPUFA, omega-3 fatty acids
Vegetables: 300–400 g/day, 3–4 portions/day
Fruits: 100–200 g, 1–2 portions/day
Legumes: beans, peas, lentils, chickpeas, soy/at least once a week
Low-fat dairy products, half a liter of milk—calcium Red meat: 350–500 g boiled/steamed/fried—per week Deep-sea fish: 100–120 g/occasion/at least two servings per week 30 g unsalted nuts, oily seeds/per day 30–45 g/day dietary fiber: whole grain, fiber-rich cereals
*To be avoided*
Simple, fast-acting sugars
Smoking Sugary and alcoholic beverages Sedentary life Salt (max. 5 g/day) Saturated fat—not exceed 10% of the total energy intake

## Data Availability

Not applicable.

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
