# Peer review of "The Importance of Nutrition in Menopause and Perimenopause—A Review"

_nutrients, 2023, doi:10.3390/nu16010027_

Round 1

Reviewer 1 Report

Comments and Suggestions for Authors

This is a clear and comprehensive review probing the importance of nutrition during menopause and peri-menopause. Overall, it is a highly innovative and detailed article that provides a full aspect on the periods of menopause and peri-menopause. This topic involves significant steps that must be taken with the help of dietitians to change the diet of those affected by menopause and peri-menopause, making it of significant interest to the readership of this publication.

I have two minor suggestions to improve this article.

In the sentence "It is only possible to change dietary habits during perimenopause with nutrition counseling and intervention," consider specifying what "it" refers to for clarity. For example, "Changing dietary habits during perimenopause is most effectively achieved through nutrition counseling and intervention."

The term "dietotherapy" is not as commonly used as "dietary therapy." Consider replacing "dietotherapy" with "dietary therapy" for better readability.

Author Response

Dear Reviewer2,

Thank you for your thorough revision of our manuscript. We greatly appreciate both your positive comments and your helpful suggestions. All the points raised are answered below, and appropriate changes have been made in the revised version of the manuscript. A final, thorough linguistic check has been provided by one of the authors with a long stay in the US.  No other alterations have been implemented.

Our replies are as follows:

Comments and Suggestions for Authors

This is a clear and comprehensive review probing the importance of nutrition during menopause and peri-menopause. Overall, it is a highly innovative and detailed article that provides a full aspect on the periods of menopause and peri-menopause. This topic involves significant steps that must be taken with the help of dietitians to change the diet of those affected by menopause and peri-menopause, making it of significant interest to the readership of this publication.

I have two minor suggestions to improve this article.

In the sentence It is only possible to change dietary habits during perimenopause with nutrition counseling and intervention, consider specifying what it refers to for clarity. For example, Changing dietary habits during perimenopause is most effectively achieved through nutrition counseling and intervention.

We appreciate the revewer’s recommendastion. The sentence has been corrected.

The term dietotherapy is not as commonly used as dietary therapy. Consider replacing dietotherapy with dietary therapy for better readability.

Thank you for your reviewer's suggestion. We have corrected the wording. 

We would like to thank our Reviewer the careful and detailed overview and useful advice.

We hope that the revised manuscript will be acceptable for publication in Your highly esteemed Journal.

Kind regards, Aliz Erdélyi, Erzsébet Pálfi and Marianna Török

Reviewer 2 Report

Comments and Suggestions for Authors

The manuscript by Erdélyi and collaborators is interesting and deals with a quite innovative topic, in line with the journal's scope.  The findings are well organized and written.

Therefore, some major revisions are recommended:

1)      Introduction: it is recommended to better explain the pathophysiological aspects that link the hormonal variations in perimenopause and menopause with obesity, mainly the deposition of visceral fat, oxidative stress, and chronic inflammation. Please, these articles should be cited (PMID: 32329636).

2)      In the text it is repeated too many times that there is a reduction in basal metabolism of even 250-300 kcal per day. Please, check and eliminate any repetitions.

3)      Table 1 is not useful. It would be more appropriate to replace it with a table summarising the main nutritional indications in menopause.

4)      The "Balanced nutrition recommendation" paragraph is not sufficiently detailed. Please, explain the differences and benefits of different dietary patterns e.g. Mediterranean Diet, VLCKD, etc, citing the most recent references in the literature.

5)      Implications of gut microbiome with menopausal woman’ s health have recently been identified. I recommend adding a paragraph on this topic, also citing this article: PMID: 36746877

6

Comments on the Quality of English Language

It is advisable to carry out a minor linguistic revision.

Author Response

Dear Reviewer2,

Thank you for your thorough revision of our manuscript. We greatly appreciate both your positive comments and your helpful suggestions. All the points raised are answered below, and appropriate changes have been made in the revised version of the manuscript. A final, thorough linguistic check has been provided by one of the authors with a long stay in the US.  No other alterations have been implemented.

Our replies are as follows:   

Comments and Suggestions for Authors

The manuscript by Erdélyi and collaborators is interesting and deals with a quite innovative topic, in line with the journal's scope.  The findings are well organized and written.

Therefore, some major revisions are recommended:

1)      Introduction: it is recommended to better explain the pathophysiological aspects that link the hormonal variations in perimenopause and menopause with obesity, mainly the deposition of visceral fat, oxidative stress, and chronic inflammation. Please, these articles should be cited (PMID: 32329636).

We appreciate the reviewer's recommendation. The requested part has been added to the introduction.

’In the female body estradiol is a hormone with extensive metabolic effects, so the absence of the cycle and the lack of periodic exposure to estrogen and progesterone cause changes not only in the target tissues of sex hormones but also apart from the reproductive system.  Estradiol  affects the central nervous system, increases food intake and basal energy consumption (basal metabolism). It increases gluconeogenesis in the liver,  having an opposite effect to insulin. In skeletal muscles estradiol increases insulin sensitivity and glucose uptake [1] and improves the function of pancreatic beta cells by increasing insulin secretion. With the onset of menopause and due to the lackingeffect of estrogen, basal metabolism of the female body decreases significantly [2]. The hunger-suppressing effect of estrogen on estrogen alpha receptors in the central nervous system is also reduced, resulting in higher calory intake [3,4]. Body composition changes parallel to the decrease in basal metabolism, as body weight increases, and fat distribution changes to an increased visceral fat mass increases [5]. The excess fat storage leads to larger adipocytes and tissue remodeling of visceral fat. Local growth factors are secreted, inducing adaptive angiogenesis, high metabolic activity, and oxygen consumption, resulting in excess production of free oxygen radicals. In response to structural damage, immune cells are recruited and accumulated in adipose tissue. The increased secretion of pro-inflammatory signaling molecules induces local and low-grade systemic inflammation [6]. This low-grade systemic inflammation plays a key role in accelerating vascular damage [7].’

2)      In the text it is repeated too many times that there is a reduction in basal metabolism of even 250-300 kcal per day. Please, check and eliminate any repetitions.

We appreciate the reviewer's recommendation. We have eliminated the repetitions.

3)      Table 1 is not useful. It would be more appropriate to replace it with a table summarising the main nutritional indications in menopause.

Thank you for your reviewer's suggestion. We have prepared a table with the most important dietary recommendations.

Table 2. The main nutritional indications in menopause

Body composition

Use body composition analysis tools to assessing nutritional status

Keep the weight in healthy range

Manage overweight, obesity: reduce the current energy by 500-700 kcal/day

Regulary physical activity

Dietary recommendations

Protein – 0.8-1-1.2 g/bwkg/day

Calcium, vitamin D, vitamin C, vitamin B

n-3 LCPUFA, omega-3 fatty acids

Vegetables: 300-400 g/day, 3-4 portions/day

Fruits: 100-200 g, 1-2 portions/day

Legumens: beans, peas, lentils, chicpeas, soy / at leat once a week

Low-fat dairy products, half a liter of milk – calcium
Red meat: 350-500 g boilded/setamed/fried – per week

Deepsea fish: 100-120 g/occacion / at least two sercvings per week
30 g unsalted nuts, oily seeds/per day

30-45 g/day dietary fiber: whole grain, fiber-rich cereals

To be avoided

Simple, fast-acting sugars

Smoking
Sugary and alcoholic beverages

Sedentary life

Salt (max. 5 g/day)
Saturated fat – not exceed 10% of the total enery intake

4)      The "Balanced nutrition recommendation" paragraph is not sufficiently detailed. Please, explain the differences and benefits of different dietary patterns e.g. Mediterranean Diet, VLCKD, etc, citing the most recent references in the literature.

Thank you as a reviewer for your suggestion, we have added a chapter on the differences and benefits of different diets.

’There are many diet which have positive effect on the chronic non-infectious diseases and the weight management, such as Mediterrian diet, Very Low Calorie Diet (VLCD-1,200 kcal/day).  Diets providing less than 1,200 kcal/day may yield micronutrient deficiencies which could affects negatively not only the nutritional status, but also the weight management outcome [8]. Therefor Low-Calories-Diet (LCD) and VLCD diets are used only in clinical practice. The balanced hypocalorie diets can be manage to individuals and may therefor have better chance for long-therm success [8]. The other popular diet is the Mediterranean diet. The Mediterranean diet is characterized from foods with anti-inflammatory and antioxidant action. There is evidence that Mediterranean diet has affect the weight management, the blood sugar control and the cardiovascular diseases [4]. At the same time the Mediterranean meal planning is not sustainable in most of the European countries. Summerising the healthy, balanced diet is achievable in long term, therefor it is preferred.’

5)      Implications of gut microbiome with menopausal woman’ s health have recently been identified. I recommend adding a paragraph on this topic, also citing this article: PMID: 36746877

Thanks to the reviewer's suggestion, we have added the requested information to the chapter. 

’8. Microbiom and menopause

Perimenopause and menopause is often associated with dysbiosis and gastrointestinal tract complaints. Estrogen levels can effect gut microbiota [9,10] and microbiome in turn can effect serum estrogen levels as certain microbes in the gut (called estrabolome) secrete beta-glucoronidase, a bacterial enzyme that deconjugates estrogens and phytoestrogens into their active forms which can be reabsorbed in the intestine and enter the bloodstream [11]. Estrabolome can be reduced by dysbiosis and consequently lead to further loss of circulating active estradiol metabolites [12]. This suggests that composition of the microbiota could determine the onset or progression of some menopause-related clinical conditions [13].

Probiotic supplementation in menopausal women suggest a favorable effect on some cardiovascular risk factors [14] through maintaining the integrity of the intestinal barrier, therefore reducing translocation of bacteria through the gut wall and decreasing systemic inflammation. Fermentation of polysaccharides and undigested proteins by some microbial strains generates SCFA (i.e. acetate, propionate, butyrate), possibly benefiting several metabolic pathways. Newest studies suggest, that supplementation with probiotics improves cardiometabolic risk factors in postmenopausal women [15]. Maintaining healthy gut biom by a balanced diet according to the needs of menopausal women might help prevent dysbiosis.’

Comments on the Quality of English Language

It is advisable to carry out a minor linguistic revision.

A final, thorough linguistic check has been provided by one of the authors with a long stay in the US.  No other alterations have been implemented.

We would like to thank our Reviewer the careful and detailed overview and useful advices.

We hope that the revised manuscript will be acceptable for publication in Your highly esteemed Journal.

Kind regards, Aliz Erdélyi, Erzsébet Pálfi and Marianna Török

  1. Foryst-Ludwig, A.; Kintscher, U. Metabolic impact of estrogen signalling through ERalpha and ERbeta. J Steroid Biochem Mol Biol 2010, 122, 74-81. doi: 10.1016/j.jsbmb.2010.06.012
  2. Poehlman, E.T. Menopause, energy expenditure, and body composition. Acta Obstet Gynecol Scand 2002, 81, 603-611. doi: 10.1034/j.1600-0412.2002.810705.x
  3. Mauvais-Jarvis, F.; Clegg, D.J.; Hevener, A.L. The role of estrogens in control of energy balance and glucose homeostasis. Endocr Rev 2013, 34, 309-338. doi: 10.1210/er.2012-1055
  4. Barrea, L.; Pugliese, G.; Laudisio, D.; Colao, A.; Savastano, S.; Muscogiuri, G. Mediterranean diet as medical prescription in menopausal women with obesity: a practical guide for nutritionists. Crit Rev Food Sci Nutr 2021, 61, 1201-1211. doi: 10.1080/10408398.2020.1755220
  5. Lovejoy, J.C.; Champagne, C.M.; de Jonge, L.; Xie, H.; Smith, S.R. Increased visceral fat and decreased energy expenditure during the menopausal transition. Int J Obes (Lond) 2008, 32, 949-958. doi: 10.1038/ijo.2008.25
  6. Crewe, C.; An, Y.A.; Scherer, P.E. The ominous triad of adipose tissue dysfunction: inflammation, fibrosis, and impaired angiogenesis. J Clin Invest 2017, 127, 74-82. doi: 10.1172/JCI88883
  7. Alexopoulos, N.; Katritsis, D.; Raggi, P. Visceral adipose tissue as a source of inflammation and promoter of atherosclerosis. Atherosclerosis 2014, 233, 104-112. doi: 10.1016/j.atherosclerosis.2013.12.023
  8. Yumuk, V.; Tsigos, C.; Fried, M.; Schindler, K.; Busetto, L.; Micic, D.; Toplak, H.; Obesity Management Task Force of the European Association for the Study of, O. European Guidelines for Obesity Management in Adults. Obes Facts 2015, 8, 402-424. doi: 10.1159/000442721
  9. Huang, G.; Xu, J.; Lefever, D.E.; Glenn, T.C.; Nagy, T.; Guo, T.L. Genistein prevention of hyperglycemia and improvement of glucose tolerance in adult non-obese diabetic mice are associated with alterations of gut microbiome and immune homeostasis. Toxicol Appl Pharmacol 2017, 332, 138-148. doi: 10.1016/j.taap.2017.04.009
  10. Flores, R.; Shi, J.; Fuhrman, B.; Xu, X.; Veenstra, T.D.; Gail, M.H.; Gajer, P.; Ravel, J.; Goedert, J.J. Fecal microbial determinants of fecal and systemic estrogens and estrogen metabolites: a cross-sectional study. J Transl Med 2012, 10, 253. doi: 10.1186/1479-5876-10-253
  11. Zimmermann, P.; Messina, N.; Mohn, W.W.; Finlay, B.B.; Curtis, N. Association between the intestinal microbiota and allergic sensitization, eczema, and asthma: A systematic review. J Allergy Clin Immunol 2019, 143, 467-485. doi: 10.1016/j.jaci.2018.09.025
  12. Plottel, C.S.; Blaser, M.J. Microbiome and malignancy. Cell Host Microbe 2011, 10, 324-335. doi: 10.1016/j.chom.2011.10.003
  13. Vieira, A.T.; Castelo, P.M.; Ribeiro, D.A.; Ferreira, C.M. Influence of Oral and Gut Microbiota in the Health of Menopausal Women. Front Microbiol 2017, 8, 1884. doi: 10.3389/fmicb.2017.01884
  14. Szulińska, M.; Łoniewski, I.; van Hemert, S.; Sobieska, M.; Bogdański, P. Dose-Dependent Effects of Multispecies Probiotic Supplementation on the Lipopolysaccharide (LPS) Level and Cardiometabolic Profile in Obese Postmenopausal Women: A 12-Week Randomized Clinical Trial. Nutrients 2018, 10. doi: 10.3390/nu10060773
  15. Barrea, L.; Verde, L.; Auriemma, R.S.; Vetrani, C.; Cataldi, M.; Frias-Toral, E.; Pugliese, G.; Camajani, E.; Savastano, S.; Colao, A.; et al. Probiotics and Prebiotics: Any Role in Menopause-Related Diseases? Curr Nutr Rep 2023, 12, 83-97. doi: 10.1007/s13668-023-00462-3